# L-Band Synthetic Aperture Radar and Its Application for Forest Parameter Estimation, 1972 to 2024: A Review

**DOI:** 10.3390/plants13172511

**Published:** 2024-09-07

**Authors:** Zilin Ye, Jiangping Long, Tingchen Zhang, Bingbing Lin, Hui Lin

**Affiliations:** 1Research Center of Forestry Remote Sensing & Information Engineering, Central South University of Forestry and Technology, Changsha 410004, China; loyzer@163.com (Z.Y.); longjiangping@csuft.edu.cn (J.L.); 20220100015@csuft.edu.cn (T.Z.); lbb15623238233@outlook.com (B.L.); 2Key Laboratory of Forestry Remote Sensing Based Big Data & Ecological Security for Hunan Province, Changsha 410004, China; 3Key Laboratory of State Forestry Administration on Forest Resources Management and Monitoring in Southern Area, Changsha 410004, China

**Keywords:** L band SAR, Citexs data, forest canopy penetration

## Abstract

Optical remote sensing can effectively capture 2-dimensional (2D) forest information, such as woodland area and percentage forest cover. However, accurately estimating forest vertical-structure relevant parameters such as height using optical images remains challenging, which leads to low accuracy of estimating forest stocks like biomass and carbon stocks. Thus, accurately obtaining vertical structure information of forests has become a significant bottleneck in the application of optical remote sensing to forestry. Microwave remote sensing such as synthetic aperture radar (SAR) and polarimetric SAR provides the capability to penetrate forest canopies with the L-band signal, and is particularly adept at capturing the vertical structure information of forests, which is an alternative ideal remote-sensing data source to overcome the aforementioned limitation. This paper utilizes the Citexs data analysis platform, along with the CNKI and PubMed databases, to investigate the advancements of applying L-band SAR technology to forest canopy penetration and structure-parameter estimation, and provides a comprehensive review based on 58 relevant articles from 1978 to 2024 in the PubMed database. The metrics, including annual publication numbers, countries/regions from which the publications come, institutions, and first authors, with the visualization of results, were utilized to identify development trends. The paper summarizes the state of the art and effectiveness of L-band SAR in addressing the estimation of forest height, moisture, and forest stocks, and also examines the penetration depth of the L-band in forests and highlights key influencing factors. This review identifies existing limitations and suggests research directions in the future and the potential of using L-band SAR technology for forest parameter estimation.

## 1. Introduction

The rapid advancement in technologies of satellites [1], unmanned aerial vehicles (UAVs) [2] and radar [3] has facilitated the acquisition of fast and precise ground information for forest resource surveys. Consequently, the reliance on remote sensing technologies in forestry has significantly increased. Remote sensing technologies now enable accurate extraction of two-dimensional (2D) forest information such as woodland area and percentage forest cover, along with other parameters. However, due to the limited penetration ability of electromagnetic waves and the complexity of forest canopies, obtaining accurate information of vertical structures remains challenging. Estimating forest canopy height and related parameters such as stock, biomass, and carbon storage continues to pose difficulties. Therefore, acquiring comprehensive information of forest vertical structures represents a major obstacle that hampers the widespread application of remote sensing technologies in forestry.

 Spaceborne microwave signals used by Synthetic Aperture Radar (SAR) and polarimetric SAR possess the capability to penetrate forest canopies. Particularly, L-band SAR signals exhibit strong forest canopy-penetration ability and sensitivity [4] to forest vertical structures. Thus, the L-band SAR offers a valuable means of acquiring extensive forest vertical-structure information across multiple frequencies and serves as an optimal remote sensing data source for addressing this “bottleneck”. However, current research findings indicate that the estimation results of forest height using L-band SAR data, combined with InSAR or PolInSAR technology, are still not satisfied. This is primarily attributed to the unclear understanding of the penetration process and response mechanism of the L-band SAR signal in forests, as well as the lack of clarity regarding its penetration depth in different types of forests. The quantitative description of transmittance and behavior patterns exhibited by the L-band SAR signal within forest canopies remains elusive, impeding the establishment of a reliable model for determining the L-band SAR signal’s penetration depth [5]. Therefore, it becomes imperative to characterize the attenuation process and penetration depth of the L-band SAR signal based on information of forest canopies obtained from air-ground cooperative microwave radiometers and LiDAR data. Additionally, interpreting the forest penetration mechanism associated with the L-band SAR signal will enable us to develop a theoretical framework and methodology for retrieving accurate estimates pertaining to forest penetration depth of the L-band, and thus effectively overcome the existing limitations related to extracting accurate information of forests through remote sensing technologies.

## 2. Literature Analysis of Forest Penetration of L-Band SAR Signal

### 2.1. Introduction of L-Band SAR Satellites

In 1978, the United States launched the first Ocean Satellite (Seasat) with the L-band SAR sensor. Then, Japan launched the JELLS-1 satellite in 1992 and the Advanced Land Observing Satellite (ALOS) in January 2006 with L-band SAR sensors and a maximum positioning accuracy of 10 m. In November 2009, the European Space Agency launched the Soil Moisture and Ocean Salinity Satellite (SMOS), which can emit L-band radiation energy to the ground and provide 9 km spatial resolution products. In January 2015, the United States launched the Active and Passive Soil Moisture Monitoring Satellite (SMAP) with an L-band SAR sensor, which can provide global moisture products with a resolution of 40 km × 40 km. In October 2018, Argentina launched the SAOCOM 1A satellite with L-band and a spatial resolution of 10 m × 10 m to 100 m × 100 m. In January 2022, China launched the Land Probe 1 Group 01 A/B satellite (LT-1A/1B), also known as the L-band differential Interferometric SAR satellite, with spatial resolutions of 3 m × 3 m, 6 m × 6 m, 12 m × 12 m, 20 m × 20 m and 30 m × 30 m. A summary of the launched satellites with L-band is shown in Table 1.

Compared with the number of other remote sensing satellites, L-band SAR satellites are very few, and even fewer are in orbit. Thus, it is relatively difficult to obtain L-band data. The emergence of LT-1 has started to change this situation.

### 2.2. Trend in Annual Publications

Based on the Citexs comprehensive literature database, this paper adopted the bibliometrics method to carry out mining of the literature big database by selecting year, country, institution, author, journal, and so on. Moreover, analyzing and visualizing the general trend and distribution in this field was conducted. Using L band SAR and forest penetration as keywords, we identified 0 papers from January 1978 to April 2024 in the CNKI database and 58 papers from January 1978 to April 2024 in the PubMed database. The average annual number of the papers published was three, and the annual numbers of the published papers are shown in Figure 1.

Figure 1 shows that the publications related to the use of L-band SAR data in the field of forest canopy penetration first appeared in 1999, indicating a relatively late start. From 1999 to 2011, there is only one SCI paper published each year, which indicates the stage of slow development. From 2012 to 2023, the annual number of the published papers increased to more than two, reaching the peak of eight in 2019. The fastest growth happened from 2012 to 2019, indicating that the research in this field was in a rising stage of rapid development, and then a decreased trend was found.

### 2.3. Publications by Country and Region

From January 1978 to April 2024, the distribution of the top 23 countries/regions in the studies of L band SAR data and forest canopy penetration in the world is shown in Figure 2. The countries/regions in which there is the largest number of publications were China (13 papers, 22.41%), then Germany (7 papers, 12.07%) and France (7 papers, 12.07%).

### 2.4. Publications by Research Institution

The top-20 national research institutions in which the authors published their papers in the field of L band SAR data applications and forest canopy penetration from January 1978 to April 2024 are shown in Figure 3. The institutions can be divided into three groups. The top group includes the Chinese Academy of Sciences, the German Aerospace Center, the Japan Aerospace Exploration Agency and the Indian Institute of Remote Sensing, in which there are three papers in this field for each institute. The second group consists of 12 research institutions, each publishing two articles in this field. The last group is composed of four institutes in which there is only one publication each. 

### 2.5. Publications by the First Author

From January 1978 to April 2024, the world’s top-30 first authors who studied the use of L band SAR data and forest canopy penetration are shown in Figure 4. The author who produced the largest number of the papers in this field is Mark L. Williams, with three papers in total. Yasser Maghsoudi, Matteo Pardini, Mohammad Javad Valadan Zoej, Marco Lavalle, Tayebe Managhebi and Masanobu Shimada tied for second place, with two publications. The authors with one publication include Liming Jiang, Om Prakash Tripathi, A.K. Milne, Junli Chen, Chadi Abdallah, J. Jomaah, Masato Hayashi, Jyotishman Deka, Michael F. Toups, Kiran Dasari, P. S. Roy, Fulong Chen, Qingwei Tong, A.C. Lee, Wei Li, Ruixia Yang, Takeo Tadono, Shashi Kumar, Lal Bihari Singha, Jean Luc Betoulle, Nicolas Baghdadi, E. Mougin, and Van Nhu Le.

### 2.6. Publication by Journal

From January 1978 to April 2024, the top 22 journals in terms of the publications dealing with L-band SAR data and forest canopy penetration are shown in Figure 5. *Remote Sensing* is the journal with the largest number of publications (10). *IEEE Transactions on Geoscience and Remote Sensing* ranked second, with three articles, and *Remote Sensing of Environment* ranked third, with two papers. There are another 19 journals with one publication.

## 3. Applications of L-Band SAR Data in Forestry

L-band SAR data have been used in forestry. Most of the data come from ALOS satellites with different polarization modes. The main applications deal with estimations of forest canopy height, moisture and forest stocks.

### 3.1. Forest Height

Kugler successfully estimated and verified the potential of L-band, P-band and X-band to estimate tree height in temperate forests by using airborne SAR sensors combined with constraints [6,7]. Zhang et al. used the improved RMoG model and ALOS-1 data to invert forest height with the estimation error reduced by 27.73% and 8.57% Asopa et al. used UAV SAR technology to estimate the tree height of tropical forests with a root mean square error (RMSE) of 4.21 m [8]. Huang et al. estimated tree height by using L-band SAR data and generated a digital terrain model (DTM) and digital surface model (DSM) validated by using UAV LiDAR data [9]. Thieu et al. proposed a new algorithm based on a mean set to increase phase, and combined it with the polarization characteristics of the VE-RVoG optimized set, developed to improve the estimation of forest height, and obtained an RMSE of 2.91 m and a correlation coefficient of 0.909 [10]. Xie et al. improved the accuracy of forest height estimation by using the new airborne PolInSAR data-processing strategy; the RMSE was significantly reduced, to 1.02 m, with a decrease of 12.86%, providing a feasible solution for forest height estimation with X-band waves [11]. Based on L-band single-baseline pooled SAR interferometric simulation data, Sui et al. proposed a standard scale optimization model suitable for various densities, successfully overcoming the failure of traditional methods in low-density regions, and effectively realized the estimation of forest height [12]. 

Moreover, Luo et al. used UAV SAR multi-baseline L-band data from the AfriSAR project to show better accuracy in forest height estimation, providing an improved method for estimating structural parameters of tropical rain forests [13]. Luo et al. conducted an estimation experiment using L-band multi-baseline fully polarized data from the AfriSAR project in the Lope Pongara pilot area and proposed a depth-based error-correction method that improved the accuracy of forest height estimation and demonstrated potential applications [14] of machine learning-interference feature prediction. Zhang et al. used the simulated L-band SAR data and combined it with the improved three-stage method to derive forest height with a significantly improved accuracy [15]. Integrating the TF-RVoG method based on time–frequency analysis and the improved single-baseline data decomposition method significantly improved the estimation accuracy of the forest canopy height model, with the RMSE dropping to 2.54 m [16]. Zang et al. combined ICESat-2 data and tree age information to propose a method to estimate the change in palm tree height in Peninsular Malaysia, and successfully produced a comprehensive map of tree height change from 2001 to 2020 [17]. The verification results showed that the estimated height was highly consistent with the actual height. Providing a spatially explicit tool with great potential for quantifying plantation stocks, Sa and Nei et al. used ALOS-2 L-band data to retrieve conifer tree height in Saihanba, Hebei Province and obtained an R^2^ of 0.67 between the SAR-based estimated and the LiDAR-based conifer tree height [18]. These studies show that L-band SAR data have great potential in forest height estimation, but the estimation accuracy cannot meet the needs of forestry production.

### 3.2. Moisture

Grant et al. used an airborne L-band microwave radiometer to study the effect of forest cover on soil water estimation in Australia and utilized the L-MEB zero-order radiative transfer model to simultaneously estimate soil water and vegetation optical depth [19]. Richaume et al. found that SAR signal was highly correlated with the optical depth, roughness and canopy density of vegetation by using SMOS for large-scale moisture estimation, and the Hr value of the spatial pattern of soil moisture was correlated with land-cover types. Their results demonstrated that the evergreen broad-leaved mixed forests and the deciduous coniferous mixed forests had higher values of Hr, ranging from 0.32 to 0.39, desert, shrub and bare soil had lower values of Hr, ranging from 0.14 to 0.16, and the Hr values of grassland and tundra cultivated land changed to between 0.20 and 0.23 [20]. Konings et al. used SMAP data and a multi-temporal dual-channel retrieval algorithm (MT-DCA) to estimate the optical depth, moisture and reflectance of large-scale vegetation [21]. Lv et al. studied the relationship between optical depth, penetration and temperature of vegetation [22]. Since the L-band is sensitive to moisture, it is often used to estimate forest moisture. Holtzman et al. used L-band radiometer towers in Red Oak forests in Massachusetts, USA, to prove that the optical depth of vegetation measured by microwave radiometers is correlated with the amount of water in vegetation [23]. These studies indicate that the optical depth of vegetation is an important index to evaluate the microwave signal transmission process in forest canopy.

### 3.3. Forest Stocks

Forest stocks include stand volume, above-ground biomass (AGB) and carbon stocks. Balzter et al. estimated changes [24] in stand volume in Thetford, United Kingdom from 1910 to 1997 using Seasat and JELLS-1 satellite data. Santoro et al. used JELLS-1 L-band SAR data to study volume of forests in Sweden, Finland and Siberia, and achieved an estimation accuracy of greater than 75% [25]. Chowdhury et al. used ALOS L-band data to estimate forest volume in Siberia and obtained an R^2^ of about 0.60 between the estimated and observed values, with an accuracy greater than 70% [26]. Santoro et al. conducted a comprehensive assessment of forest volume by using L-band ALOS data from 2006 to 2011; they found that HH-polarized SAR data had a good estimation effect, and obtained an error of less than 30% when the area was larger than 20 hectares [27,28]. Thiel et al. used ALOS data to estimate forest volume in central Siberia; the R^2^ between the estimated and measured values reached 0.58, and the estimation accuracy was greater than 70% [29]. Christian Thiel et al. employed ALOS PALSAR L-band to estimate forest volume in central Siberia, and demonstrated that HV backscattering achieved a slightly higher accuracy than HH. They also found that the simple inversion method, coupled with multi-temporal SAR images, performed well in feature correction, providing a feasibility study for forest resource estimation in this region. Santoro et al. compared the forest volume-estimation potential of SAR data in X-, C- and L-bands, and demonstrated that L-band led to the highest accuracy, with a relative error of 31.3% [30]. Zhang et al. used ALOS-2 (PALSAR-2) data to estimate forest volume in Huangfengqiao Forest Farm, Hunan Province, and obtained an R^2^ of 0.61 [31].

Sandra Englhart et al. used X-L band SAR data to estimate forest AGB in Indonesian Borneo, and the results showed that the X-L band was suitable for the estimation when AGB was low, with an R^2^ of 0.53 [32]. Oliver Cartus et al. used ALOS PALSAR data to carry out regional scale mapping of forest biomass in northeastern United States. They combined SAR data and optical remote-sensing calibration models, and the results showed that the accuracy and performance of the method was superior to the results from using SAR data alone, which are dependent on imaging conditions [33]. Peregon and Yamagata used L-band SAR data to carry out AGB estimation on deciduous forests in Western Siberia, and obtained an R^2^ of 0.72 with an estimation accuracy of 85% [34]. Rahman et al. analyzed different observation models of ALOS PALSAR data, and conducted a regression analysis and estimation of natural forests in southeast Bangladesh [35]. Chaparro et al. studied the use of C- and X-band vegetation optical depth to estimate forest biomass and carbon balance, and concluded that vegetation optical depth from the L-band provided more accurate information because the penetration of microwaves through the canopy is higher at longer wavelengths and lower frequencies [36]. Berninger et al. used L-band and C-band SAR data for large-scale AGB monitoring, providing important information for accurately portraying forest loss [37,38]. Liu et al. compared airborne P-band and L-band TomoSAR measurements of the canopy-height model (CHM) and AGB over a tropical forest in Lope, Gabon, and found that the results of the CHM did not significantly differ, while the P-band was more sensitive than the L-band in the estimation. Maciej J. Soja et al. used P-band SAR data to estimate AGB in tropical forests and obtained an accuracy of 80%, based on the field measurements from 141 plots [39,40]. 

Ji et al. studied the sensitivity of L-band SAR backscattering with respect to forests with conditions of different mean canopy densities, different mean tree height, and different mean diameter at breast height (DBH), and found that the way of backscattering affected the improvement in biomass estimation accuracy. Hernandez-Stefanoni et al. improved the AGB map of tropical arid forests by integrating LiDAR, ALOS PALSAR, and climate data, and reduced the relative error of biomass estimation by 12.2% [41]. Zhang et al. used L-band ALOS data to estimate the AGB of Chinese fir forests in Huangfengqiao Forest Farm, Hunan Province, by extracting multiple rotation thresholds, and realized an estimation accuracy of 77.5% [42]. Ni et al. estimated the biomass of deciduous forests in mountainous areas with three-dimensional (3D) data and found that the season had a significant impact on the estimation results [43]. These studies show that the L-band has the potential to estimate forest stock and biomass, but the accuracy of estimation results is generally lower than 80%, due to inaccurate information of forest vertical structures.

## 4. Penetration of L-Band Signal and Its Influencing Factors

### 4.1. Penetration of L-Band Signal

Dal pointed out that microwave signals can penetrate vegetation and that the elevation bias caused by penetration is not exactly equal to the penetration depth, then proposing a penetration model [44]. Pardini and Papathanassiou carried out a forest canopy-penetration experiment using L- and P-band data, and their results showed that SAR penetration ability was closely related to band and canopy structure [45]. In Brazivella, Congo, Toochi et al. investigated the penetration capacity of six bands, including the K-band (1 cm), X-band (3 cm), C-band (5.6 cm), S-band (10 cm), L-band (23 cm), and P-band (75 cm), and further confirmed that the penetration of short wavelengths (X, K) was low and that the penetration depth was also dependent on the vertical structures of the forests [46]. Reginald R. Muskett progressively buried mesh reflectors underground in the Alaskan tundra, USA, and quantified the depth of L-band penetration into the soil [47]. Schlund et al. conducted penetration-depth and compensation experiments for temperate forests using an X-band based on LiDAR data. The estimated RMSE was less than 1 m, indicating great potential [48]. Teubner et al. explored the relationship between vegetation optical depth and gross primary production (GPP), and found that, overall, GPP was negatively correlated with vegetation optical depth in predominantly occurring both wet and dry areas, and that the correlation was similar to higher SIF [49]. 

Tanase et al. evaluated the effectiveness of C-band, L-band and P-band SAR sensors in Romanian coniferous forests using simulation models, and their results showed that different bands had different sensitivity to vegetation characteristics and disturbances. The authors emphasized the need for the comprehensive use of multi-band, dense time series and different types of sensors to compensate for the limitations of a single frequency and acquisition time [50]. Chaparro et al. quantified the contribution of ACD proportional vegetation optical depth/enhanced vegetation index signals, and their results confirmed an enhancement compared to higher frequency bands, indicating that the penetration depths of all bands were reduced in the densest forests. The 34% and 30% of variance could be explained with the proportional decrease in C- and X-band vegetation optical depth [51], respectively. Colliander et al. used airborne L-band SAR data to carry out a two-year forest-soil-moisture experiment under forest canopies in the northeastern United States [52]. Singh et al. found that the penetration depth of microwave signals into the ground varied significantly with the available content of soil moisture, with longer wavelengths having the stronger ability to penetrate the soil; however, the penetration capacity would be reduced as soil moisture content increased [53]. Liu et al. used ALOS L-band data to conduct penetration studies in extremely arid desert areas, and their results showed that the penetration depth of the L-band reached 2.98 m [54]. Qi et al. conducted an additional reference-height error analysis for baseline calibration based on a distributed Target DEM in TwinSAR-L in the arid region of eastern Xinjiang. Their results showed errors of 1.295 m and 1.39 m, respectively, which seriously reduced the product quality [55]. 

Wang et al. selected AGB, the leaf area index (LAI), and the normalized difference vegetation index (NDVI) to optimize effective scattering albedo (ω), surface roughness, and for estimation (VODini). When LAI was greater than 20.76 and NDVI larger than 0.83, the results were significantly improved, especially for dense vegetation [56]. Zhu et al. used ALOS data and a deep-learning algorithm to study penetration depth in desert areas, and found that the maximum penetration depth of the L-band reached 2.84 m, and that the penetration was also related to scattering coefficient, dielectric constant, surface roughness and mineral composition [57]. The simulation of Bai et al. showed the vegetation optical depth increased linearly with the decrease in LAI, while the results were similar to those from the satellite-based L-band, C-band and X-band. Their sensitivity tests indicated that polarization dependence become more pronounced at higher frequencies [58]. Olivares-Cabello et al., through unsupervised classification analysis on a global scale, found that the L-band is suitable for monitoring dense canopies, while X-band and LCX vegetation optical depths are more suitable for sparse tree canopy, savannas and grasslands [59]. Schmidt et al. analyzed the relationship between live fuel coverage and vegetation optical depth through random forest regression, using multi-band datasets and soil-moisture—marine-salinity sensors, providing important guidance for selecting suitable wavelengths for specific applications and algorithm development [60]. Baur et al. studied the attenuation characteristics of L-, C- and X-bands with respect to the conditions of different land-cover types, and concluded that shrub had a high transient peak value, while forest canopy had a low value [61]. In the areas with strong seasonal rainfall, the seasonal amplitude was greater in the C-band than in the L-band. The penetration characteristics of the L-band in various conditions and its comparison with the C-band and X-band are summarized in Table 2.

### 4.2. Influencing Factors in L-Band Penetration

Singh et al. used the functional relationship between incidence angle and ground penetration depth and found that penetration ability was related to incidence angle, wavelength, and soil characteristics, including water moisture content and structural composition [62]. The longer the wavelength, the stronger the penetration ability, but the penetration ability decreased with soil depth. Ji et al. studied the sensitivity of different canopy densities, mean stand height, and mean diameter at breast height to L-band backscattering [63]. They concluded that canopy density had a greater effect on L-band backscattering than average height and mean diameter at breast height. HV is more sensitive to forest structural parameters than HH, also depending on the tree species. Richaume et al. found that the L-band signal was not only related to the optical depth of vegetation, but also to the canopy roughness of tree species and canopy density. Zhu et al. found that the L-band was related to ground roughness and even mineral composition, such as hematite reducing the penetration depth. These studies indicate that there are many factors affecting the penetration of the L-band in forests, mainly including incidence angle, polarization mode, forest canopy density, tree species (canopy roughness), crown height (age structure) and moisture. Moreover, slope, season, meteorological factors, forest LAI, leaf direction and so on, also affect the penetration effect.

## 5. Future Development

### 5.1. Integration of L-Band SAR Data and the Tomography Algorithm

Cazcarra-Bes et al. used TomoSAR technology to process L-band forest data obtained by monitoring at different times, extracted the distribution and spatial patterns of forests by a compressed sensing approach, and further utilized two complementary search methods to find the local maximum value and reconstruct the spatial patterns, so as to realize the inference of forest structure information and the assessment of the effect of delineating forests [64]. Minh et al. found that in the forests with a height of above 30 m and biomass up to 500 t/hm^2^, the strong ground return of P-band can be seen in the tomography images [65]. Tello et al. used L-band SAR data in Trauenstein, Germany, to provide forest vertical-structure information, coupled with high spatial- and temporal-resolution images; they developed TomoSAR technology and reconstructed 3D models, and the results were verified based on airborne LiDAR data [66]. This experiment opened the door for 3D-based forest monitoring. Moussawi et al. compared P-band and L-band TomoSAR profiles, using the Land Vegetation Ice Sensor (LVIS), and discrete-return LiDAR to monitor and estimate tropical forest-structure parameters [67]. The extracted radar reflectance yielded RMSE values of 3.02 m and 3.68 m for P-band and L-band, respectively, and the corresponding determination coefficients were 0.95 and 0.93. Pardini et al. reviewed the features of L-band TomoSAR reconstruction and discussed the unique ability of reconstructing radar reflectance using TomoSAR to reveal the 3D structure and temporal changes of forests [68]. The authors emphasized the importance of penetration sensitivity to vegetation elements.

### 5.2. Integration of L-Band and P-Band 

Minh D et al. believed that the further development of integrating the L-band and P-band would make it possible to use TomoSAR technology to more accurately extract forest vertical structures, which would not only solve the problem of forest classification, but also provide strong support for the next generation of Earth Explorer BIOMASS mission [69]. Lope, Gabon, Liu et al. compared the CHM and AGB models of a tropical forest obtained by an airborne P-band and L-band synthetic (SAR) Tomosar tomography [70]. In the forests located in Paracou, French Guiana and South America, Ngo et al. analyzed the applications of airborne P-band TomoSAR and LiDAR, showing that both could directly lead to high-resolution surface, height, and profile models. The results demonstrated that airborne-based products had higher quality due to stronger penetration. For the forest of an average height of 30 m at Paracou, a RMSE of less than 5 m for tree height estimation was obtained [71]. Moreover, Chuang et al. proposed a robust TomoSAR imaging procedure to obtain local high-resolution L-band images of forests for the areas of interest [72].

The aforementioned studies demonstrate the strong penetration of L-band SAR signal in forests, and reveal the key factors that affect the penetration ability of the L-band SAR signal in forests. However, the studies did not explain the attenuation mechanism of the SAR signal in forest canopies and also did not account for the response process of microwave transmittance and penetration depth. In the estimation of forest height, the improvement in estimation accuracy was mainly achieved by developing algorithms and enhancing computation performance. Obtaining the results, to some extent, was fortuitous.

In addition, the existing experiments focus mainly on temperate and boreal forests, and rarely take place in subtropical forests. Compared with the temperate and boreal forests, the structures of subtropical forests are more complex. Generally, subtropical forests often consist of three layers including tree, shrub, and grass, and have higher canopy closure and moisture inside the forests, which will lead to stronger impacts on the penetration of signals. Therefore, the results from the temperate and boreal forests will be less effective, and their applicability is limited in subtropical forests.

L-band SAR data provide the potential for advancing forestry remote sensing technologies by exploring forest vertical structures from the canopy surface to the interior of forests and from 2D- to 3D-model reconstruction l [73], and are also critical in realizing the technologies of TomoSAR and penetrating remote sensing in the future [74].

## 6. Conclusions and Discussion

Through bibliometric analysis of the related literature, this paper reviews the application status of L-band SAR data and their penetration in forests. The main conclusions are drawn as follows:

(1) The number of the publications is related to the availability of L-band SAR data. The earliest launched L-band SAR satellite took place in 1978; thus, this paper deals with the activities from 1972 to 2024. The first publication reviewed appeared in 1999, which indicates that, compared with other remote sensing technologies, the development of L-band-related technology started late. From 1999 to 2011, there was only one publication coming out each year, and less attention was paid. From 2012 to 2024, the number of the relevant publications increased rapidly and L-band SAR data (including airborne data) began to appear widely.

(2) The lack of L-band SAR data impedes their application. Compared with a large number of other remote-sensing data, L-band SAR data are relatively less available (see Table 1). Some satellites have ceased operation and others have just appeared recently, with data such as SAOCOM and LT. There has been no relevant literature found. The existing publications mainly deal with the use of Japan’s JERS and ALOS data. With the emergence of SAOCOM, LT and UAV data, this situation should be changed in the near future.

(3) The existing reports concentrate mainly on the temperate and boreal forests, while there have rarely been relevant L-band studies conducted in the vast subtropical forests. This situation is mainly related to the countries/regions in which the relevant research was carried out. The countries/regions with most of the existing studies are mainly distributed in the temperate and boreal zones, including China (mainly in the Chinese Academy of Sciences), Germany and France.

(4) The state of the art in applications of L-band SAR data: the existing studies show that the L-band signal has strong penetration in snow, soil moisture, water–land boundaries, vegetation, vegetation moisture, and in the ocean and desert underground. The applications of L-band SA data in forestry focus mainly on the estimation of forest height, moisture and forest stocks. The factors affecting the penetration of the L-band in forests mainly include forest-canopy closure, tree species, crown height and moisture. Slope, seasonality, meteorological features, LAI and leaf direction also have influence on the penetration.

(5) Directions for future efforts: the existing literature deals mainly with applied studies of L-band data, while theoretical research studies hardly exit. There are a lack of studies related to the mechanism of L-band signal working in various forests and the interactions between the L-band signal and the forests, especially subtropical forests. For example, the penetration capacity and attenuation process and characteristics of the L-band signal in various forests are unknown. The penetration depth, the transmission rate and response process of the L-band in different forests are also unclear. Moreover, there have rarely been reports dealing with the compensation mechanism of using L-band SAR data to estimate forest height. The gaps that exist currently imply the importance of carrying out the research on the mechanism of forest penetration of the L-band SAR signal. In addition, the integration of L-band and new technologies, such as P-band and chromatographic SAR data, will provide a technical way to improve forest height estimation.

## Figures and Tables

**Figure 1 plants-13-02511-f001:**
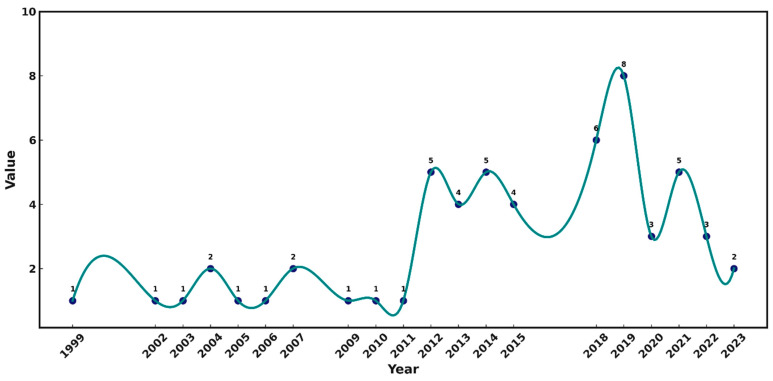
SCI published papers from 1978 to 2024 dealing with L-band SAR data and related to forest penetration.

**Figure 2 plants-13-02511-f002:**
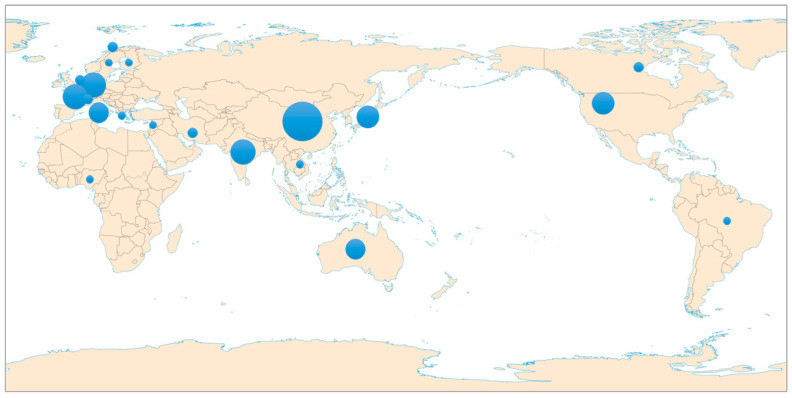
The distribution of the SCI publications related to the use of L-band SAR data and forest canopy penetration.

**Figure 3 plants-13-02511-f003:**
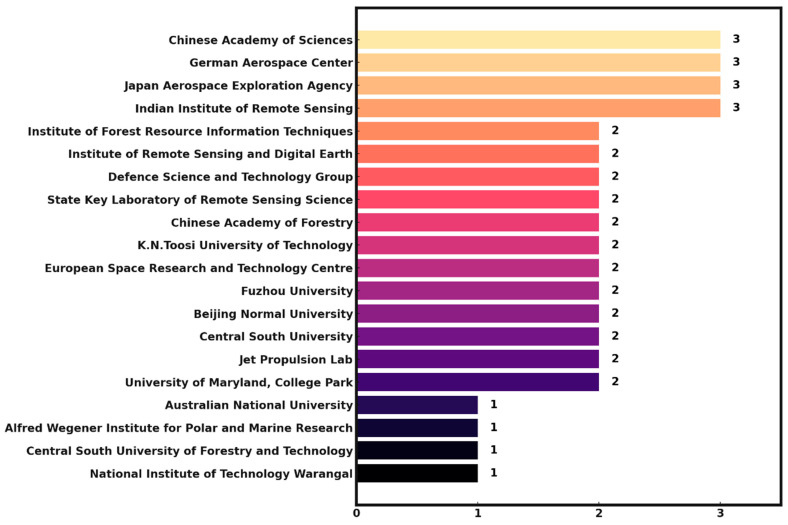
Top-20 research institutions that published SCI papers dealing with L-band SAR data and forest canopy penetration.

**Figure 4 plants-13-02511-f004:**
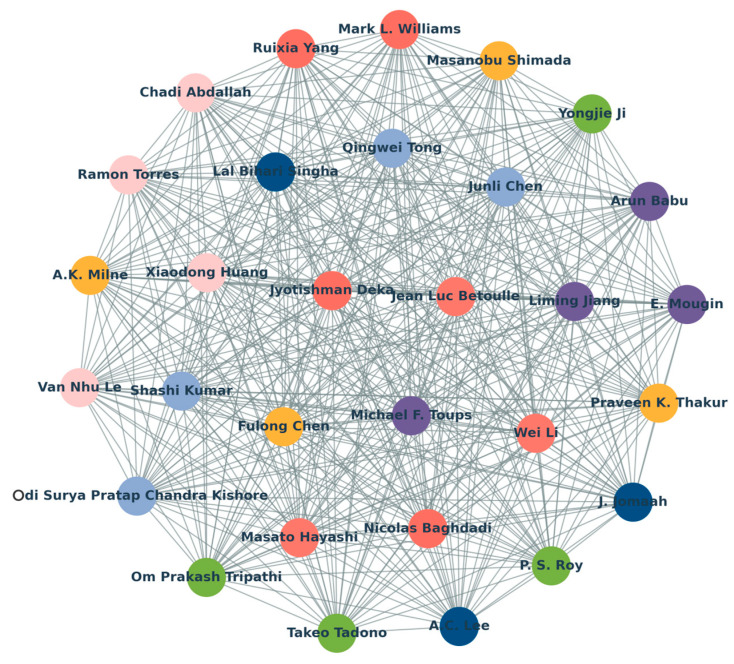
The-top 30 first authors who published SCI articles in the field of using L-band SAR data and dealing with forest canopy penetration.

**Figure 5 plants-13-02511-f005:**
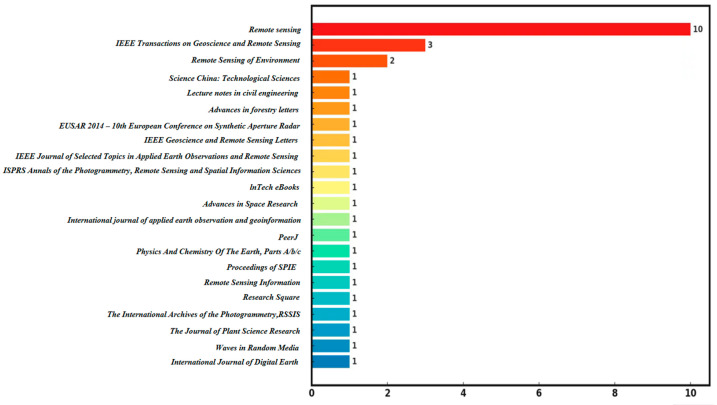
Publications by journal in the field of using L-band SAR data and dealing with forest canopy penetration.

**Table 1 plants-13-02511-t001:** SAR satellites with L band in the world.

Serial Number	Launch Time	Country or Region	Satellite Name
1	February 1978	Global Positioning System	NAVSTAR GPS
2	June 1978	American Seasat satellite	Seasat
3	February 1992	Japan Earth Resource Satellite	JERS-1
4	January 2006	Japan Advanced Land Observing Satellite	ALOS/ALOS-2
5	November 2009	European Space Agency Soil Moisture and Ocean Salinity Satellite	SMOS
6	January 2015	United States Active and Passive Soil Moisture Monitoring Satellite	SMAP
7	October 2018	Argentine Microwave Observation Satellite 1A	SAOCOM-1A
8	September 2019	China Yunhai-1 02 satellite	Yunhai-1 02
9	August 2020	Argentine Microwave Observation Satellite 1B	SAOCOM-1B
10	January 2022	China Landexplorer-1 Group 01A satellite	LT-1A
11	January 2022	China Landexplorer-1 Group 01 B satellite	LT-1B

**Table 2 plants-13-02511-t002:** Penetration characteristics of L-band in various conditions and its comparison with C-band and X-band (Note: √√—very good; √—good; *—not good).

Observed Object	L-Band	C-Band	X-Band
Sea Ice	*	√	√√
Snow (type and thickened layer)	√√	√√	√√
Soil moisture	√√	√√	√√
Soil roughness	√	√√	√√
Soils	√√	√	*
Water-land boundaries	√√	√	√√
Vegetation	√√	√√	√√
Vegetation moisture	√√	√√	√√
Ocean	√√	√√	*
Geological structure, structure	*	√	√
Desert underground	√√	√	*

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
