# Peer review of "L-Band Synthetic Aperture Radar and Its Application for Forest Parameter Estimation, 1972 to 2024: A Review"

_plants, 2024, doi:10.3390/plants13172511_

Round 1

Reviewer 1 Report

Comments and Suggestions for Authors

I was fascinated by these advances in research mentioned in the work. Obviously their work is more of a bibliography than real discoveries from the use that is cited. Great job on research and development. 

Author Response

We thank the reviewer for your positive comments and useful suggestions. All of them were addressed to improve the quality of the revised manuscript.

Q1: I was fascinated by these advances in research mentioned in the work. Obviously their work is more of a bibliography than real discoveries from the use that is cited. Great job on research and development. 

The author’s answer: We express our gratitude for your invaluable guidance. In the future, we will prioritize research and development efforts to validate our findings through practical implementation."

Reviewer 2 Report

Comments and Suggestions for Authors

This paper offers a comprehensive and detailed review of the advancements in L-band SAR technology for forest parameter estimation from 1978 to 2024. It effectively highlights the limitations of optical remote sensing in capturing forest vertical structures and demonstrate the unique capabilities of L-band SAR in overcoming these challenges. By employing a robust methodology that utilizes the Citexs data analysis platform along with CNKI and PubMed databases, it presents a detailed bibliometric analysis that identifies trends in publication, key contributing countries, institutions, and leading authors in the field. This approach provides a clear picture of the current state and progression of research, enhancing the paper’s clarity and impact.

The data and results presented are clear and well-illustrated through figures and tables, such as the listing of L-band SAR satellites and the trend of annual publications. These visual aids effectively convey the findings and contribute to the paper’s comprehensibility. The authors delve into specific applications of L-band SAR data, including the estimation of forest height, moisture, and stocks, and provide detailed examples from various studies that showcase the practical utility of this technology. By referencing specific research works, such as those by Kugler in 2016 and Asopa et al. in 2020, the paper illustrates both the effectiveness and challenges of using L-band SAR.

Importantly, the paper discussed the limitations of current research, such as the unclear understanding of L-band SAR signal penetration and the lack of data availability. It also proposes future research directions, including the integration of L-band with other technologies like P-band and TomoSAR to improve accuracy in forest parameter estimation. The authors emphasize the need for more theoretical research to understand the mechanisms of L-band SAR signal interaction with different forest types, identifying gaps in existing studies, particularly the need for research in subtropical forests, which are more complex and less studied than temperate and boreal forests.

The paper offers novel insights into the potential of L-band SAR technology for forestry applications, highlighting areas where this technology could overcome current limitations in remote sensing. It identifies the significant impact of forest canopy closure, tree species, crown height, and moisture on L-band penetration, as well as other influencing factors like slope, seasonality, and meteorological features. The authors suggest that the integration of L-band and new technologies, such as P-band and chromatographic SAR data, will be crucial for future advancements.

Overall, the paper is well-organized, with clear sections that guide the reader through the introduction, literature analysis, applications, and future research directions. The writing is clear and concise, making complex technical information accessible to a broad audience. This paper is a significant contribution to the field of forestry remote sensing, providing a valuable resource for researchers and practitioners. 

However, there are a few drawbacks worth noting:

1. Limited Geographical Scope: The focus is primarily on temperate and boreal forests, with insufficient attention to subtropical forests, which may have different and more complex structures.

2. Scarcity of Data: The review highlights the limited availability of L-band SAR data, particularly from newer satellites, which may constrain the applicability and generalizability of the findings.

3. Lack of Theoretical Depth: There is an insufficient exploration of the theoretical mechanisms underlying L-band SAR signal interaction with forest canopies, which is crucial for enhancing the accuracy of parameter estimation.

To address these drawbacks, I encourage the authors to consider the following modifications or provide feedback:

 1. Expand the Geographical Scope: Include more studies and examples related to subtropical forests to provide a more comprehensive overview of the applicability of L-band SAR across different forest types.

 2. Discuss Data Availability Solutions: Explore potential solutions or future prospects for increasing the availability of L-band SAR data, especially from newer satellites.

 3. Enhance Theoretical Discussion: Include a more detailed exploration of the theoretical mechanisms of L-band SAR signal interaction with forest canopies, possibly by integrating more recent studies or theoretical models.

These modifications will strengthen the paper and provide a more holistic view of the current state and future potential of L-band SAR technology in forest parameter estimation.

Author Response

We thank the reviewer for your positive comments and useful suggestions. All of them were addressed to improve the quality of the revised manuscript.

Q1: Limited Geographical Scope: The focus is primarily on temperate and boreal forests, with insufficient attention to subtropical forests, which may have different and more complex structures.

The author’s answer: Literature search reveals that the utilization of L-band for studies in the subtropical zone has been relatively limited, with a majority of research concentrated in the temperate, cold temperate, and cold zones. Only a few papers have explored this band in the tropical and subtropical regions. The inadequate research outcomes observed in the subtropical zone may be attributed to several factors. Additionally, geographical distribution of research institutions primarily located in temperate zones (as depicted in Figure 2) could also contribute to this trend.

Q2: Scarcity of Data: The review highlights the limited availability of L-band SAR data, particularly from newer satellites, which may constrain the applicability and generalizability of the findings.

The author’s answer: The research on the penetration capability of L-band in forest has been actively pursued over the past decade, primarily utilizing Japanese satellite data. In the last five years, several countries have successfully deployed and operated five L-band satellites (refer to Table 1); however, no significant findings have been obtained.

Q3: Lack of Theoretical Depth: There is an insufficient exploration of the theoretical mechanisms underlying L-band SAR signal interaction with forest canopies, which is crucial for enhancing the accuracy of parameter estimation.

The author’s answer: Estimating forest height using remote sensing technology has posed a challenging problem in the field of forestry remote sensing. The advent of L-band offers a potential solution to this issue. However, the underlying mechanism governing the interaction between L-band and forests remains unclear, particularly regarding variations in L-band attenuation among different tree species and canopy densities, as well as the impact of forest height (age) and internal humidity on L-band attenuation. These factors directly influence the penetration depth of L-band within forests and subsequently affect the accuracy of forest height estimation, warranting further investigation.

Reviewer 3 Report

Comments and Suggestions for Authors

            This manuscript provides a comprehensive review of L-Band applications on retrieving forest parameters. It shows L-Band has been widely applied in tree height, forest stocks and soil moisture. I have some general comments.

1. Can L-Band be used to estimate tree canopy cover? Canopy cover is an important factor and is used to minimize the background effect when scaling the data. I saw some studies using C-band to retrieve canopy cover? However, it did not mentioned in this study. Can you discuss this and add it to the manuscript. Also, machine learning is a hot toic in SAR area, did you find some studies using L-BAND + ML to retrieve forest factors? I think it is worth discussing. The below manuscript would be helpful for discussion.

Gao, S., Castellazzi, P., Vervoort, R.W. and Doody, T.M., 2021. Fine scale mapping of fractional tree canopy cover to support river basin management. Hydrological Processes, 35(4), p.e14156.

2. Again, machine learning would be very relevant to this topic, it needs to discuss the future applications or limitations of using L-Band.

3. Line 112: delete non-English characters.

Author Response

We thank the reviewer for your positive comments and useful suggestions. All of them were addressed to improve the quality of the revised manuscript.

Q1: Can L-Band be used to estimate tree canopy cover? Canopy cover is an important factor and is used to minimize the background effect when scaling the data. I saw some studies using C-band to retrieve canopy cover? However, it did not mentioned in this study. Can you discuss this and add it to the manuscript. Also, machine learning is a hot toic in SAR area, did you find some studies using L-BAND + ML to retrieve forest factors? I think it is worth discussing. The below manuscript would be helpful for discussion.

Gao, S., Castellazzi, P., Vervoort, R.W. and Doody, T.M., 2021. Fine scale mapping of fractional tree canopy cover to support river basin management. Hydrological Processes, 35(4), p.e14156.

The author’s answer: Thank you for your valuable comments and documentation. However, this review primarily provides a comprehensive overview of the current state and advancements in L-band research pertaining to forest penetration, specifically addressing three key aspects: forest height estimation, humidity assessment, and forest reserve mapping. The existing limitations are identified along with suggested directions for future efforts, while also highlighting the potential prospects for L-band technology in penetrating forests.

Lidar is indeed very accurate in observing forest parameters, such as ground three-dimensional laser data and airborne laser data, but they have in common that the cost is too high, the observation range is small, and the production is difficult to promote.

Q2: Again, machine learning would be very relevant to this topic, it needs to discuss the future applications or limitations of using L-Band.

The author’s answer: Thank you for your valuable comments. Many of the research papers mentioned in my review have used machine learning algorithms to improve accuracy in their studies

Q3: Line 112: delete non-English characters.

The author’s answer: 112 lines of characters have been modified

Round 2

Reviewer 3 Report

Comments and Suggestions for Authors

no further comments